# Transcriptome and Metabolome Analysis Revealed the Freezing Resistance Mechanism in 60-Year-Old Overwintering *Camellia sinensis*

**DOI:** 10.3390/biology10100996

**Published:** 2021-10-03

**Authors:** Hui Wu, Zixian Wu, Yuanheng Wang, Jie Ding, Yalin Zheng, Heng Tang, Long Yang

**Affiliations:** Agricultural Big-Data Research Center and College of Plant Protection, Shandong Agricultural University, Taian 271018, China; 18864805709@163.com (H.W.); wzx886302@163.com (Z.W.); wyh931529059@163.com (Y.W.); dddinn@163.com (J.D.); 18864805679@163.com (Y.Z.); tangheng333@163.com (H.T.)

**Keywords:** transcriptome, metabolome, freezing resistance, substances, pathways

## Abstract

**Simple Summary:**

The freezing stress during overwintering brings great challenges to the normal growth of *Camellia sinensis*. The current research on *C. sinensis* mainly focuses on cold resistance, but less on freezing resistance. In the present study, the transcriptome and metabolome of *C. sinensis* under freezing stress were studied. Results showed that *Pyr/PYL-PP2C-SnRK2* played a critical role in the signal transduction of freezing stress. Three metabolic pathways including phenylpropanoid biosynthesis, flavone and flavonol biosynthesis, and flavonoid biosynthesis contributed to the freezing resistance of *C. sinensis*. This study provides substantial insights for the breeding of *C. sinensis*.

**Abstract:**

Freezing stress in winter is the biggest obstacle to the survival of *C. sinensis* in mid-latitude and high-latitude areas, which has a great impact on the yield, quality, and even life of *C. sinensis* every year. In this study, transcriptome and metabolome were used to clarify the freezing resistance mechanism of 60-year-old natural overwintering *C. sinensis* under freezing stress. Next, 3880 DEGs and 353 DAMs were obtained. The enrichment analysis showed that pathways of MAPK and ABA played a key role in the signal transduction of freezing stress, and *Pyr/PYL-PP2C-SnRK2* in the ABA pathway promoted stomatal closure. Then, the water holding capacity and the freezing resistance of *C. sinensis* were improved. The pathway analysis showed that DEGs and DAMs were significantly enriched and up-regulated in the three-related pathways of phenylpropanoid biosynthesis, flavone and flavonol biosynthesis, and flavonoid biosynthesis. In addition, the carbohydrate and fatty acid synthesis pathways also had a significant enrichment, and the synthesis of these substances facilitated the freezing resistance. These results are of great significance to elucidate the freezing resistance mechanism and the freezing resistance breeding of *C. sinensis*.

## 1. Introduction

*Camellia sinensis* is a perennial evergreen flowering plant of the genus Camellia in the Camellia family, which has the characteristics of loving a warm and humid climate [1,2,3]. In addition, it can grow regularly at 10–35 °C but becomes nearly dead at −10 °C. Shandong province is a northernmost tea growing region in China and has the highest latitude in tea growing regions of the world except for the Mediterranean climate regions [4]. In the 1960s, Chinese “South Tea being introduced to the North” was one of the most successful examples of introduction in the world, and Tai’an, Shandong, was one of the successful sites for the first trial planting.

Freezing stress in winter is the biggest obstacle to the survival of *C. sinensis* in mid-latitude and high-latitude areas and an important factor causing crop yield reduction and even death [5,6,7,8]. When plants are subjected to freezing stress, photosynthesis will be inhibited and cell structure destroyed, resulting in oxidative damage, metabolic disorder, cell damage, and other phenomena [9,10,11]. Simultaneously, plants will express a large number of resistance genes and metabolites when subjected to freezing stress [12].

In order to clarify the mechanism of freezing resistance in plants, scientists carried out a lot of research in molecular biology and bioinformatics [13,14]. Under freezing stress, *Fritillaria cirrhosa* would accumulate high H_2_O_2_ and activate antioxidant systems such as catalase, anthocyanins, phenols, SOD, and ascorbic acid peroxidase [15]. After the relief of freezing stress, the photosynthesis of *F. cirrhosa* would return to the normal level. Studies on *Triticum aestivum* leaves under freezing stress showed that *T. aestivum* could improve frost resistance by the accumulation of sucrose and the coordination of salicylic acid and carbon [16]. Transcriptional sequencing of *Brassica napus* under freezing stress revealed 3905 DEGs, of which the up-regulated DEGs were mainly in hormone signal transductions, energy metabolism, and resistance related gene families [17]. With the development of bioinformatics, more and more research has been done to explore the mechanism of plant responses to abiotic stresses by combining transcriptome with metabolome [18,19]. Transcriptome and metabolome of *Lycium barbarum* under salt stress were analyzed to obtain 1396 DEGs and 71 DAMs [20]. The pathway analysis showed that the metabolism of flavonoids could improve the salt tolerance of *L. barbarum*. The flavonoid and phenolic compounds produced by phenylpropanoid biosynthesis, flavone and flavonol biosynthesis, and flavonoid biosynthesis pathways had a positive response on abiotic stresses [21]. Therefore, it is of great significance to probe these metabolic pathways to understand the mechanism of plant resistance. The research on transcriptome and metabolome of *C. sinensis* mainly focuses on the cold treated greenhouse materials. The results showed that there was a close relationship between cold stress and drought stress, and *ICE* (inducer of CBF expression) and *HSP* (heat shock proteins) gene families had a positive response under cold stress in *C. sinensis* [22,23].

*C. sinensis* is one of the most important cash crops in the world [24,25]. Although it has the characteristics of a perennial, a large number of *C. sinensis* trees are frozen to death during overwintering every year and need to be replanted in the second year, which consumes a lot of labor and financial resources. In this study, transcriptome and metabolome were used to examine the freezing resistance mechanism of 60-year-old *C. sinensis* trees under natural freezing stress, and to clarify the gene regulation of *C. sinensis* trees under the extreme temperature of −14 °C. Furthermore, metabolic pathways and the metabolic state of *C. sinensis* trees under freezing stress were analyzed. From the gene and metabolite level, the reason why these *C. sinensis* trees can spend 60 winters in Shandong province was explored, which laid a foundation for the excavation of the freezing resistance genes and the cultivation of the freezing resistance varieties of *C. sinensis*.

## 2. Materials and Methods

### 2.1. Plant Material and Sampling

The experiment was carried out in the 60-year-old overwintering *C. sinensis* garden (36°13′9″ N, 117°3′16″ E) on Mount Tai (Tai ’an, Shandong province, China). The first sampling (freezing stress) took place on 30 December 2020, with a minimum temperature of −14 °C and an average temperature of −10 °C as the lowest temperature of the year. On 30 March 2021, a second sample (control) was taken of the tea plants at an average temperature of 15 °C. Tea trees with a good growth status were selected for the first sampling, the tea trees were marked after sampling, and the same tea tree was selected for the second sampling. These samples were immediately frozen in liquid nitrogen and stored at −80 °C for subsequent sequencing [26]. The experiment was divided into the freezing stress group (FS) and control check group (CK). A total of 9 samples were sampled from each group, of which 3 samples (FS_1, FS_2, FS_3, CK_1, CK_2, CK_3) were sequenced for transcriptome and 6 samples (FS-1, FS-2, FS-3, FS-4, FS-5, FS-6, CK-1, CK-2, CK-3, CK-4, CK-5, CK-6) were analyzed for metabolome.

### 2.2. Transcriptome Sequencing, Identification, and Analysis of Differentially Expressed Genes (DEGs)

Using the Truseq RNA (RNAprep Pure Plant Kit, Poly-saccharides&Polyphenolics-rich, centrifugal column, Beijing, China) sample preparation kit, the RNA sample was prepared according to the manufacturer’s protocol [27]. RNA quality was evaluated by agarose gel electrophoresis and OD260/230 ratio, and the cDNA library was constructed after the samples were qualified [28]. The library was qualified by an Agilent 2100 bioanalyzer (Agilent Technologies, Palo Alto, CA, USA) and quantified by Qubit and qPCR [29]. Then, the constructed cDNA library was sequenced with Illumina HiSeq 2500 and transformed into original Sequenced Reads using BCL2FASTQ [30,31]. The raw data were evaluated with FASTQC, followed by filtering (removing the containing adapter and low-quality reads) raw data with FASTP, and then again using FASTQC to perform quality control [32,33]. Hisat2 was used to map the clean data to “shuchazao” reference genome on 10 May 2021 (http://tpia.teaplant.org/download.html, accessed on 10 May 2021) and calculate mapping information [34]. The Stringtie software was used to assemble transcripts and predict expression levels using tea transcriptome data [35]. R language was used to analyze the correlation between tea transcripts and then draw the heat map [36].

The differentially expressed genes (DEGs) of log2fc ≥ 2 and *p*-value ≤ 0.05 under freezing stress were screened by lemma, and the R package was used to make volcano plots. DEGs were annotated and enriched by Gene Ontology (GO) (http://www.geneontology.org, accessed on 10 May 2021) and Kyoto Encyclopedia of Genes and Genomes (KEGG) databases (www.kegg.jp/kegg/pathway.html, accessed on 10 May 2021), respectively, to obtain the function and pathway results of DEGs, and the results were visualized by R packages [37,38].

### 2.3. Untargeted Metabolomics Analysis

Tea leaves were ground with liquid nitrogen (100 mg), homogenized with 100 µL, and suspended with precooled 100% methanol (−20 °C), and then swirled fully. Samples were kept at −20 °C for 60 min and centrifuged at 4 °C for 14,000× *g* for 15 min. Then, supernatants were transferred to a fresh microcentrifuge tube and dried in a vacuum evaporator. After drying, the metabolites were redissolved with 80% methanol and analyzed by the HPLC-MS/MS platform [39]. Next, the data were extracted using Compound Discover V3.1 (CD) software, including noise filtering, retention time alignment, mass spectrometry peak extraction, compound mapping between samples, compound identification, gap filling, and background subtracting. Meanwhile, compounds were identified using the mzCloud and ChemSpider database [40,41,42]. Finally, based on the quality control (QC) sample, it was filtered, and data standardization and normalization were achieved by the Constant SUM and Auto Scaling algorithm [43,44].

### 2.4. Metabolome Data Processing and Analysis

The standardized tea metabolomic data were analyzed by a principal component analysis (PCA) and orthogonal partial least squares discriminant analysis (OPLS-DA), and visualized as scatter plots [45,46]. Differentially accumulated metabolites (DAMs) between the FS group and CK group were determined according to t-test *p* value < 0.05 and VIP > 1. Finally, the DAMs metabolic pathway was annotated and enriched based on the KEGG pathway database, and the results were visualized using R language.

## 3. Results

### 3.1. C. sinensis Transcriptome Sequencing Results and Data Analyses

The raw data of 251 G were obtained by Illumina high-throughput sequencing, and the clean data of 247 G were obtained by filtering, and the clean rate of each group of data was greater than 97.71% (Appendix A). Q20 were all greater than 98%, and Q30 were all greater than 93%. The mapping ratio of clean data to reference genome was 89% to 90%, in which both uniquely mapped reads and multiple mapped reads were around 84% and 16%, respectively. The average transcript number of the FS group (70,616) was smaller than the CK group (73,232). In addition, the exon total length, average transcript length, and N50 length (without intron) of the FS group were also smaller than the CK group.

### 3.2. Transcriptional Characteristics of C. sinensis Response to Freezing Stress

The correlation analysis showed that the r values of transcriptome data within both FS and CK were greater than 0.9, and the r values between groups were about 0.85 (Figure 1A). Compared with CK, a total of 3880 DEGs were obtained under freezing stress, of which 1876 DEGs were up-regulated and 2004 DEGs were down-regulated (Figure 1B). The GO enrichment analysis of DEGs showed that DEGs had a large amount of enrichment in biological processes, cell components, and molecular functions (Figure 1C). In terms of biological processes, the number of expression genes in the metabolic process and the cellular process all exceeded 277, and the number of expression genes in the single-organism process was 172. Gene expression was all highly enriched in cell components including the membrane, membrane part, cell, cell part, and organelle. Gene expression was mainly concentrated in the molecular functions including catalytic activity and binding, and the number of gene expressions was close to 400. In addition, under freezing stress, *C. sinensis* also enriched significantly in the immune system process, responses to stimulus, antioxidant activities, and other stress reactions.

### 3.3. Pathway Analysis in Response to Freezing Stress

The KEGG enrichment analysis suggested that DEGs were mainly enriched in photosynthesis, sugar and flavonoid metabolism, and signal transduction (Figure 2A). Furthermore, the *MAPK* pathway played an essential role. The analysis of the *MAPK* signaling pathway under freezing stress showed that *PYR/PYL-PP2C-SnRK2* had a positive response to abscisic acid (ABA) signal transduction (Figure 2B). The physiological changes of stress adaptation and stomatal closure were responded to by ABA signal transduction. A large number of DEGs was assigned to the ABA signal transduction pathway, among which PYR/PYL-PP2C-SnRK2 was very significant. In the *PYR/PYL* gene family, five genes were significantly up-regulated, and *CSS0047272* and *CSS0017736* were the most significant up-regulated with log(FC) value greater than four. Moreover, the *PP2C* gene family and *SnRK2* gene family had one significantly down-regulated gene and one significantly up-regulated gene, respectively (Figure 2C).

### 3.4. Metabolic Characteristics of C. sinensis Response to Freezing Stress

A total of 10543 DAMs were obtained through the non-targeted metabolomic analysis of *C. sinensis* leaves under freezing stress, and 353 metabolites were assigned to 171 KEGG functional categories (Appendix A and Figure 3C). Freezing stress and CK had significantly different effects on the expression of metabolites, and great similarity within samples of the same condition. To further overview the distinctions between *C. sinensis* leaves under freezing stress and CK, the multivariate analysis was performed on metabolic data. A principal component analysis (PCA) of the 10,543 DAMs showed that the two treatments were significantly differentiated for the first component (79.3% for PC1) (Figure 3A). The OPLS-DA of the 10,543 DAMs showed a clearer distinction between the FS and CK groups of samples (Figure 3B). The R2X, R2Y, and Q2 of OPLS-DA model were 0.775, 1, and 0.996, respectively, with high reliability. A total of 353 metabolites were mapped to 170 pathways by KEGG annotation, in which DAMs were mainly enriched in diterpenoid biosynthesis, biosynthesis of secondary metabolites, flavone and flavonol biosynthesis, and other amino acid metabolic processes (Figure 3D).

### 3.5. Correlation Analysis of Gene Expression and Metabolite Levels

The KEGG annotation of DEGs and DAMs of freezing-stressed *C. sinensis* leaves showed that both sets of the data were significantly enriched in phenylpropanoid biosynthesis, flavone and flavonol biosynthesis, and flavonoid biosynthesis (Appendix A, Figure 4A,B). Indeed, 32 DAMs were intensively enriched in the three connected pathways, of which 31 DAMs were up-regulated and only one down-regulated. All the DAMs in flavone and flavonol biosynthesis and flavonoid biosynthesis pathways were up-regulated. Moreover, 48 DEGs engaged in the pathway, and most of them were up-regulated. Alcohols and aldehydes produced in phenylpropanoid biosynthesis were the initial compounds for lignin formation [47]. Up-regulated Epigallocatechin was the most abundant flavonoidin in green tea and was an important potential source of antioxidants [48]. In addition, a number of flavonoid metabolites (luteolin, luteoloside, Gallocatechin, etc.) increased and were significantly associated with plant stress in the flavone and flavonol biosynthesis and flavonoid biosynthesis pathways [49].

## 4. Discussion

*C. sinensis* is a warm-loving crop, and its yield, quality, and even life will be seriously affected by the freezing injury. The technology of combining transcriptome and metabolome has been widely used in humans, animals, and plants, and played an essential role in the study of abiotic stresses in plants. In this study, the freezing-resistance mechanism of 60-year-old *C. sinensis* trees under natural freezing stress was studied using transcriptome and metabolome.

In transcriptome sequencing, the transcript number of the FS samples was smaller than that of the CK samples, and the number of down-regulated DEGs was higher than the number of up-regulated DEGs. Many of these down-regulated genes were associated with the normal development and physiological metabolism of *C. sinensis*, suggesting that at 10 °C *C. sinensis* recovered from the dormancy state to normal vegetative growth. The enrichment of DEGs was observed in the photosynthesis–antenna proteins pathway. Photosystem II (PSII) was one of the most sensitive components of photosynthesis. In addition to the reactive oxygen species (ROS) produced by freezing stress, the light-harvesting complex also produced ROS which could damage PSII and inhibit the photosynthesis of *C. sinensis* under freezing stress.

According to the GO enrichment analysis, the membrane was the most enriched cell structure of DEGs. Moreover, the biological processes (e.g., response to stimulus) and molecular functions (e.g., antioxidant activity and signal transducer activity) of GO enrichment were related to abiotic stresses and inseparable from the membrane. Cell membrane played a critical role in the response of *C. sinensis* to freezing stress. Through the analysis of DEGs pathways, the *MAPK* pathway and ABA signal transduction processes had a positive response to freezing stress in *C. sinensis* leaves. A large number of genes of *PYR/PYL-PP2C-SnRK2*, the core module of ABA signal transduction, were up-regulated to transmit the signal of freezing stress to promote *C. sinensis* and produce stress adaptation. *Pyr/PYL-PP2C-SnRK2* contributed to stomata closure to improve the water holding capacity and freezing resistance of *C. sinensis*. The *MAPK* pathway process and ABA signal transduction mediated the information transmission in the freezing stress of *C. sinensis*. In addition, ABA can be used as a dormant hormone to promote leaf shedding [50]. *C. sinensis* metabolic pathways were also enriched in many material syntheses such as carbohydrates and flavonoids. Carbohydrates can provide energy for plants, and many flavonoids are associated with stress tolerance. These substances can also increase the concentration of cell fluid to improve freezing resistance [51]. *C. sinensis* could resist freezing stress through complex stress processes composed of stress signal transduction, substance synthesis, and physiological changes.

The pathway enrichment analysis of DAMs and DEGs exhibited positive responses to freezing stress in phenylpropanoid biosynthesis, flavone and flavonol biosynthesis, and flavonoid biosynthesis pathways. DEGs regulated three pathways to produce 31 significantly up-regulated metabolites and to resist freezing stress in *C. sinensis*. Phenylalanine and cinnamic acid in the phenylpropanoid biosynthesis pathway were precursor substances for lignin synthesis, and the expression of regulatory genes *CSS0018870* and *CSS0021474* were significantly up-regulated, which were related to lignin synthesis. Lignin, the main component of the cell wall, can promote the transport of minerals in plants and improve the ability of water retention, and act as the first barrier for plants to resist adverse external environments to cope with stress [52]. *C. sinensis* responded to freezing stress by regulating the synthesis of substances through phenylpropanoid biosynthesis pathways. All DAMs in the flavone and flavonol biosynthesis and flavonoid biosynthesis pathways were significantly up-regulated. Flavonoids are one of the plant’s most biologically active secondary metabolites and have significant antioxidant activity [53]. Luteolin 7-O-glucoside, which mainly exists in vacuoles, can enhance the adaptive ability of plants [54]. Myricetin, a molecule with six hydroxyl groups, has antioxidant properties [55]. In addition, the reduction of flavonoids will reduce the freezing resistance of plants [56,57]. Flavonoids played an essential role in the freezing stress of *C. sinensis*.

## 5. Conclusions

In conclusion, *PYR/PYL-PP2C-SnRK2* showed a positive response in the signal transduction of freezing stress in *C. sinensis*. In addition, the phenylpropanoid biosynthesis, flavone and flavonol biosynthesis, and flavonoid biosyn-thesis pathways regulated by genes encoding enzymes might play a key role. In this experiment, large numbers of DEGs, metabolites, and key pathways were observed from *C. sinensis* under freezing stress. It is worth studying further how the freezing stress signals are conducted and how these significantly related candidate genes and metabolites regulate the freezing resistance of *C. sinensis*. The results laid a foundation for improving the freezing resistance of *C. sinensis* by genetic engineering.

## Figures and Tables

**Figure 1 biology-10-00996-f001:**
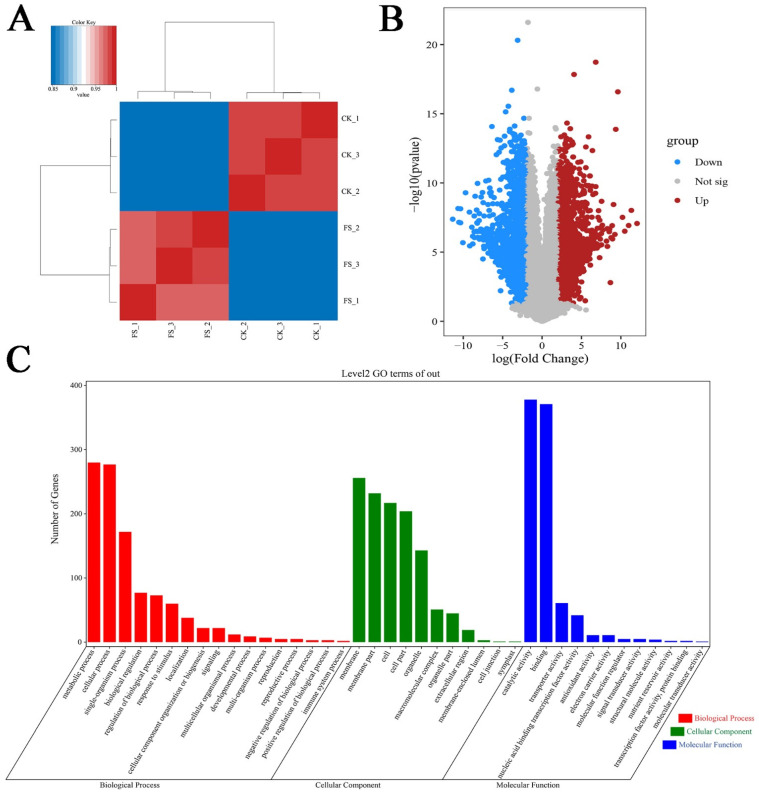
Correlation analyses between the gene expression of samples under control and freezing stress conditions. (**A**) Correlations between different samples. (**B**) Volcano plot of DEGs identified under freezing stress. Red dots represent up-regulated DEGs and blue dots represent down-regulated DEGs. (**C**) GO annotation results for the respective DEGs.

**Figure 2 biology-10-00996-f002:**
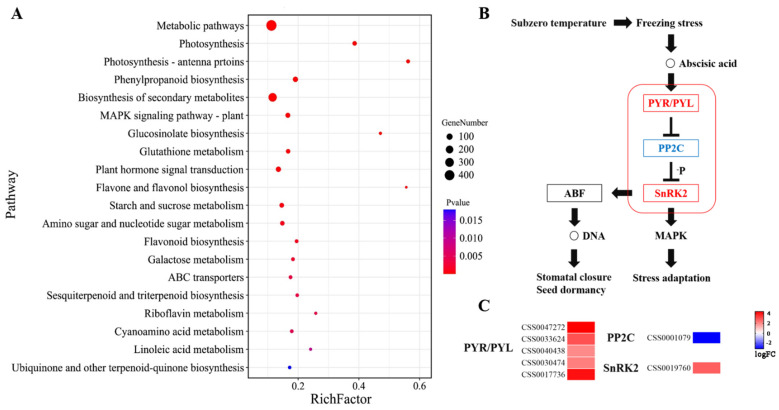
The pathway analysis of DEGs. (**A**) Bubble plot of the results of DEGs KEGG annotation. (**B**) Regulation of ABA signal transduction pathway under freezing stress. (**C**) Gene expression of ABA signal transduction pathway after freezing stress.

**Figure 3 biology-10-00996-f003:**
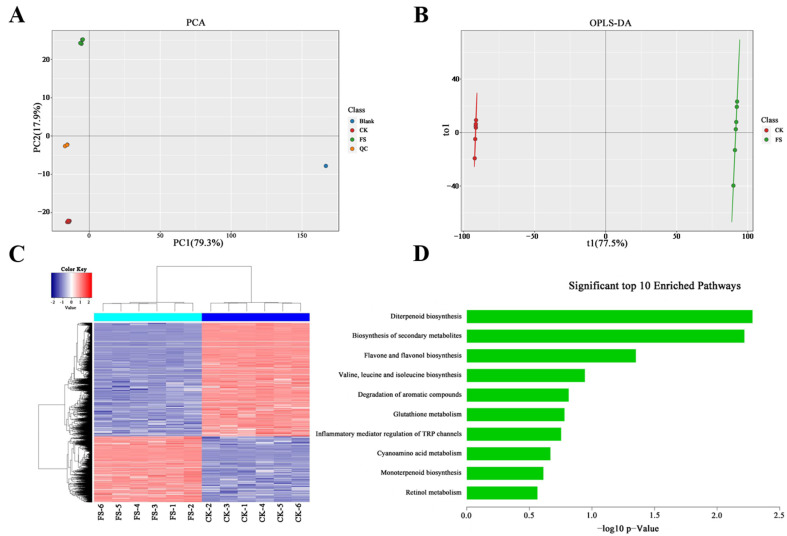
The quality and functional analysis of *C. sinensis* metabolome data. (**A**) The scatter plot of PCA scores of *C. sinensis* samples under freezing stress and CK. (**B**) OPLS-DA modeling of *C. sinensis* samples under freezing stress and CK. (**C**) The DAMs cluster heat map between samples. (**D**) The enrichment analysis of KEGG pathway top 10 for DAMs in *C. sinensis*.

**Figure 4 biology-10-00996-f004:**
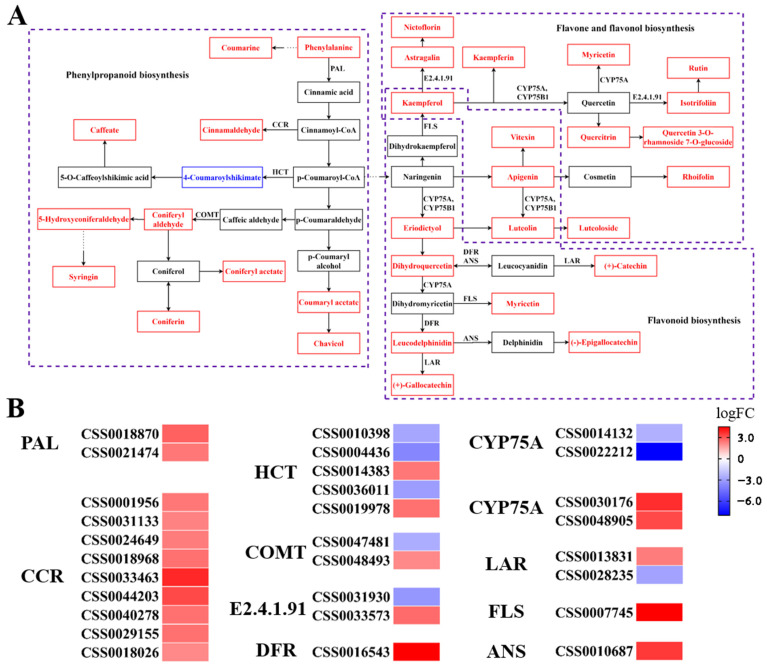
Gene expression regulates substance metabolism. (**A**) The enrichment analysis of regulatory genes encoding enzymes and DAMs in phenylpropanoid biosynthesis, flavone and flavonol biosynthesis, and flavonoid biosynthesis pathways. The box represents metabolites, red represents up-regulated metabolites, and blue represents down-regulated metabolites. Between the metabolites are regulatory respective genes encoding enzymes. (**B**) Expression of genes involved in phenylpropanoid biosynthesis, flavone and flavonol biosynthesis, and flavonoid biosynthesis pathways of freezing stress.

## Data Availability

Not applicable.

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
