# Peer review of "Transcriptome and Metabolome Analysis Revealed the Freezing Resistance Mechanism in 60-Year-Old Overwintering Camellia sinensis"

_biology, 2021, doi:10.3390/biology10100996_

Round 1
Reviewer 1 Report
In the present study, the authors investigate the transcriptomics and metabolomics changes produced by freezing stress in 60-year-old Over-3 wintering Camellia sinensis, suggesting that an activation of Pyr/PYL-PP2C-SnRK2 pathway played a critical role in the signal transduction of freezing stress. To support this hypothesis, the authors show a differential regulation of few genes involved in the Pyr/PYL-PP2C-SnRK2 in the ABA pathway that contribute to enhance freezing tolerance in C. sinensis. Indeed, the regulation of this pathway may participate to alleviate the stress, as has been reported in other species. However, and considering that the authors carried out a whole transcriptomic analysis, it would be interesting to show the information about the gene expression of other components of the pathway. For instance, Arabidopsis thaliana and rice contain 80 and 78 PP2Cs genes, some of which are up-regulated by cold stress (Xue et al., 2008). Following this, for example, it has been showed that antisense inhibition of a PP2C accelerates cold acclimation in Arabidopsis (Tähtiharju and Palva, 2001). In this sense, it would be interesting to identify the Arabidopsis (or other plant species as rice) homologous gene of C. sinensis identified in the transcriptomic analysis in response to freezing to obtain a better idea of its potential role. More importantly, it is necessary to show the regulation of all the DEGs (the authors did not include this information) and more specifically those related to the Pyr/PYL-PP2C-SnRK2 pathway, since the authors based their conclusion in the expression of few genes. For instance, the authors affirm (lines 170-172) that “the analysis of the MAPK signaling pathway under freezing stress showed that PYR/PYL-PP2C-SnRK2 had a positive response to abscisic acid (ABA) signal transduction” but they do not show the expression of MAPK codifying genes or protein accumulation. Similarly, they conclude that the activation of the PYR/PYL-PP2C-SnRK2 pathway conducts to a stomata closure that improves freezing tolerance in C. sinensis but they do not show any physiological information about it.
Another example that point out the need to include the whole transcriptomic data is the high number of genes that the authors indicate that are down-regulated in C. sinensis during the freezing stress as response to the “normal” physiological acclimation of the plant (Discussion section). Thus, the authors need to include this data and provide the necessary information to suggest this response.
In addition to the transcriptomic analysis, the authors performed a metabolomics profile to elucidate the role of different metabolites in response to freezing stress in C. sinensis. Among these metabolites, the authors found that flavone, flavonol and flavonoid biosynthesis are up-regulated under freezing conditions, which is not surprising since flavonoids are been previously described as important molecules to cope with freezing stress (Korn et al., 2008; Schulz et al., 2016).
It is interesting to know the transcriptomic and metabolomic changes occur in C. sinensis (a non-model plant) during freezing stress, which responses since to be similar to other plants responses (which is also not surprising). In this sense, the novelty of this study lies in the plant species chosen and less in the response itself. The mechanisms are not deeply studied and there are not physiological results that support the hypothesis made base on the transcriptomic analysis (stomata closure…), only assumptions.
On another hand, it is really surprising to observe that there is not even one bibliography reference during the discussion. The authors hypothesize about the responses of C. sinensis according to their results but they do not refer these results to any previous work.
Finally, it is necessary to review some sentences and C. sinensis is not well writing several times (including in the title of the article).
Author Response
All the authors thank the Reviewer #1 for giving us this comment. All the authors fully agree with what the Reviewer #1 are pointing out. We really appreciate your help and patience. We have efforted to correct the mistakes and improve the English of the manuscript. In addition, we have improved references to background and further descriptions of the methods. And we changed the conclusions to make them more credible. For the detailed response, please see below.
In the present study, the authors investigate the transcriptomics and metabolomics changes produced by freezing stress in 60-year-old Over-3 wintering Camellia sinensis, suggesting that an activation of Pyr/PYL-PP2C-SnRK2 pathway played a critical role in the signal transduction of freezing stress. To support this hypothesis, the authors show a differential regulation of few genes involved in the Pyr/PYL-PP2C-SnRK2 in the ABA pathway that contribute to enhance freezing tolerance in C. sinensis. Indeed, the regulation of this pathway may participate to alleviate the stress, as has been reported in other species. However, and considering that the authors carried out a whole transcriptomic analysis, it would be interesting to show the information about the gene expression of other components of the pathway. For instance, Arabidopsis thaliana and rice contain 80 and 78 PP2Cs genes, some of which are up-regulated by cold stress (Xue et al., 2008). Following this, for example, it has been showed that antisense inhibition of a PP2C accelerates cold acclimation in Arabidopsis (Tähtiharju and Palva, 2001). In this sense, it would be interesting to identify the Arabidopsis (or other plant species as rice) homologous gene of C. sinensis identified in the transcriptomic analysis in response to freezing to obtain a better idea of its potential role. More importantly, it is necessary to show the regulation of all the DEGs (the authors did not include this information) and more specifically those related to the Pyr/PYL-PP2C-SnRK2pathway, since the authors based their conclusion in the expression of few genes. For instance, the authors affirm (lines 170-172) that “the analysis of the MAPK signaling pathway under freezing stress showed that PYR/PYL-PP2C-SnRK2 had a positive response to abscisic acid (ABA) signal transduction” but they do not show the expression of MAPKcodifying genes or protein accumulation. Similarly, they conclude that the activation of thePYR/PYL-PP2C-SnRK2 pathway conducts to a stomata closure that improves freezing tolerance in C. sinensis but they do not show any physiological information about it.
Reply: Thanks for your nice suggestion. All the authors fully agree with what the Reviewer #1 are pointing out. We really appreciate your help and patience. We identified seven DEGs (CSS0047272, CSS0033624, CSS0040438, CSS0030474, CSS0017736, CSS0001079 and CSS0019760) from three families of Pyr/PYL, PP2C and SnRK2. The seven DEGs showed positive response to freezing stress. In addition, we added all the DEGs (Table S2).
“The physiological changes of stress adaptation and stomatal closure responded by ABA signal transduction in the C.sinensis leaf under freezing stress. (lines 172-174)” changed to “The physiological changes of stress adaptation and stomatal closure responded by ABA signal transduction.”
“In conclusion, the signals sensed by the freezing stress receptors of C.sinensis under freezing stress were transduced and transmitted through hormones, enzymes, etc., and the C.sinensis that received the signals were metabolized by lignin, flavonoids, carbohy-drates and other substances to improve stress resistance and promote the body to produce stomata closure to resist freezing stress.” changed to “In conclusion, the signals sensed by the freezing stress receptors of C.sinensis under freezing stress were transduced and transmitted through enzymes, etc., and the C.sinensis that received the signals were metabolized by lignin, flavonoids, carbohydrates and other substances. In addition, stomatal closure might be promoted by ABA signaling to resist freezing stress.”
Another example that point out the need to include the whole transcriptomic data is the high number of genes that the authors indicate that are down-regulated in C. sinensis during the freezing stress as response to the “normal” physiological acclimation of the plant (Discussion section). Thus, the authors need to include this data and provide the necessary information to suggest this response.
Reply: Thanks a lot for your suggestion. All the authors fully agree with what the Reviewer #1 are pointing out. We added the whole transcriptomic data and provided details of all DEGs in Table S2.
In addition to the transcriptomic analysis, the authors performed a metabolomics profile to elucidate the role of different metabolites in response to freezing stress in C. sinensis. Among these metabolites, the authors found that flavone, flavonol and flavonoid biosynthesis are up-regulated under freezing conditions, which is not surprising since flavonoids are been previously described as important molecules to cope with freezing stress (Korn et al., 2008; Schulz et al., 2016).
Reply: Thanks a lot for your suggestion. KEGG annotation of DEGs and DAMs of freezing-stressed C.sinensis leaves showed that both sets of the data were significantly enriched in phenylpropanoid biosynthesis, flavone and flavonol biosynthesis and flavonoid biosynthesis (Figures 4A and 4B). 32 DAMs were intensively enriched in the three connection pathways, of which 31 DAMs were up-regulated and only 1 down-regulated. All the DAMs in flavone and flavonol biosynthesis and flavonoid biosynthesis pathways were upregulated. Moreover, 48 DEGs engaged in the pathway, and most of them were up-regulated. Flavone, flavonol and flavonoid biosynthesis were significantly up-regulated in freezing-stressed C.sinensis leaves, which played an essential role.
It is interesting to know the transcriptomic and metabolomic changes occur in C. sinensis (a non-model plant) during freezing stress, which responses since to be similar to other plants responses (which is also not surprising). In this sense, the novelty of this study lies in the plant species chosen and less in the response itself. The mechanisms are not deeply studied and there are not physiological results that support the hypothesis made base on the transcriptomic analysis (stomata closure…), only assumptions.
Reply: Thanks very much for this comment. In this study, transcriptome and metabolome analysis were performed for the first time using natural overwintering C.sinensis. 3,880 DEGs and 353 DAMs were obtained. The enrichment analysis showed that pathways of MAPK and ABA played a key role in the signal transduction of freezing stress, and Pyr/PYL-PP2C-SnRK2 in the ABA pathway might cause stomatal closure. DAMs were significantly enriched and up-regulated in the three-related pathways of phenylpropanoid biosynthesis, flavone and flavonol biosynthesis and flavonoid biosynthesis. Significant DEGs (CSS0047272, CSS0033624, CSS0040438, CSS0030474, CSS0017736, CSS0001079 and CSS0019760) and important pathways related to C.sinensis freezing resistance were obtained.
On another hand, it is really surprising to observe that there is not even one bibliography reference during the discussion. The authors hypothesize about the responses of C. sinensisaccording to their results but they do not refer these results to any previous work.
Reply: Thanks for your nice suggestion. We included references to previous research in the discussion.
References
[1] Wang Y, Xiong F, Nong S, Liao J, Xing A, Shen Q, Ma Y, Fang W, Zhu X. Effects of nitric oxide on the GABA, polyamines, and proline in tea (Camellia sinensis) roots under cold stress. Sci Rep. 2020 Jul 22;10(1):12240.
[2] Liu Q, Luo L, Zheng L. Lignins: Biosynthesis and Biological Functions in Plants. Int J Mol Sci. 2018 Jan 24;19(2):335.
[3] Wang W, Li Y, Dang P, Zhao S, Lai D, Zhou L. Rice Secondary Metabolites: Structures, Roles, Biosynthesis, and Metabolic Regulation. Molecules. 2018 Nov 27;23(12):3098.
[4] Böttner L, Grabe V, Gablenz S, Böhme N, Appenroth KJ, Gershenzon J, Huber M. Differential localization of flavonoid glucosides in an aquatic plant implicates different functions under abiotic stress. Plant Cell Environ. 2021 Mar;44(3):900-914.
[5] Knez Hrnčič M, Ivanovski M, Cör D, Knez Ž. Chia Seeds (Salvia hispanica L.): An Overview-Phytochemical Profile, Isolation Methods, and Application. Molecules. 2019 Dec 18;25(1):11.
Finally, it is necessary to review some sentences and C. sinensis is not well writing several times (including in the title of the article).
Reply: Thanks very much for this comment. We have efforted to correct the mistakes and improve the English of the manuscript. And the usage of C.sinensis in the title and text have been changed. Partial changes are as follows:
“Correlation analyses between samples and gene expression. (A) The analysis of correlation and difference between samples within and between groups. (B) Volcano plot of DEGs under freezing stress. Red dots represent up-regulated DEGs and blue dots represent down-regulated DEGs. (C) GO annotation results for DEGs.” changed to “Improve to: Correlation analyses between the gene expression of samples under control and freezing stress conditions. (A) Correlations between different samples. (B) Volcano plot of DEGs identified under freezing stress. Red dots represent up-regulated DEGs and blue dots represent down-regulated DEGs. (C) GO annotation results for the respective DEGs.”
“The ABA signal transduction had a large number of DEGs” changed to “A large number of DEGs was assigned to the ABA signal transduction pathway”.
“The results of bubble plot of DEGs KEGG annotation” changed to “Bubble plot of the results of DEGs KEGG annotation”.
“there was a great similarity within the groups” changed to “great similarity within samples of the same condition”.
“Gene expression of phenylpropanoid biosynthesis, flavone and flavonol biosynthesis and flavonoid biosynthesis pathways of freezing stress.” changed to “Expression of genes involved in phenylpropanoid biosynthesis, flavone and flavonol biosynthesis and flavonoid biosynthesis pathways of freezing stress.”
“Integrative Analysis of Transcriptome and Metabolome Reveals the Freezing Resistance Mechanism in 60-year-old Overwintering Camellia Sinensis” changed to “Integrative Analysis of Transcriptome and Metabolome Reveals the Freezing Resistance Mechanism in 60-year-old Overwintering Camellia Sinensis”.
“The physiological changes of stress adaptation and stomatal closure responded by ABA signal transduction in the C.sinensis leaf under freezing stress” changed to “The physiological changes of stress adaptation and stomatal closure responded by ABA signal transduction”.
Reviewer 2 Report
The authors investigated 60-year old tea plants on transcriptomic and metabolomics level to get insights in the underlying freezing tolerance mechanisms. They mainly identified biosynthesis of secondary metabolites, as e.g. flavonoids, as main driver of freezing tolerance.
The language in the whole manuscript has to be intensively revised, especially in the discussion. Several sentences contain inaccuracies generated by language uncertainties.
Samples were only taken in winter at subzero temperatures and in spring at warmer temperatures, but a sampling in fall before winter would have been desirable to analyze the initial status before freezing.
Sampling was only done in one year, even that for experiments under natural conditions a sampling over two to three years is necessary to detect general patterns even under changing weather conditions. Additionally, only a very low amount of samples was analyzed with three for transcriptomics and six for metabolomics.
In principle, the manuscript does not describe an integrative analysis (see headline) of transcriptomic and metabolomics data as for such an analysis data should be correlated together.
The sampling needs to be described in more detail in the methods (time of the day, which leaves, how many leaves etc.).
In the results section it is sometimes unclear if the authors just explain an assignment to certain pathways or really a statistical enrichment analysis, as often mentioned.
In the discussion, a lot of references are missing for the statement taken, for examples see detailed comments below. The discussion is in general quite poor, e.g. no references for the role of flavonoids in freezing tolerance are cited.
In the conclusion, statements were drawn on hormones and enzymes even that the authors did not measure them.
Fig. 5 is not very helpful, as it is too general. The first part was not investigated at all by the authors, fatty acid was not mentioned at all and lignin only shortly in the whole manuscript and the physiological response mentioned in the green box was not shown at all.
A detailed author contribution is missing at the end of the manuscript.
Further comments:
L25: please rephrase as pathways cannot have a concentration
L42: When plants are subjected to freezing stress, photosynthesis will be inhibited and cell structure destroyed, resulting in
L49: After being freeze stressed, the photosynthesis of F. cirrhosa would return to the normal level – the statement should be checked, as photosynthesis is inhibited after freezing stress as stated in L42
L58: „were analyzed“ as the metabolome cannot be sequenced
L60-62: Flavonoids are not known for their regulatory effects.
L72-74: delete “for 60 years” as this implied that study was done over 60 years
L74: here “gene regulation and expression” describe similar things
L75/76: Only the metabolic status, not the pathways were analyzed. Only conclusions were drawn on metabolic pathways.
L77: protein level was not analyzed
L94: investigated for metabolome analysis
L97-99: The methods should be written in the respective order, first preparation of RNA samples and the libraries, then sequencing.
L106/107: check sentence
L118: 100 l – this amount seems to be very high, please check
L127: quality control
L145-146: Sentence should be checked and an explanation ahould be given for different samples quality.
L156-161: No statistical gene enrichment analysis was done, so avoid the term enrichment, rather mention that DEG were assigned to the respective GO categories
Figure 1. Improve to: Correlation analyses between the gene expression of samples under control and freezing stress conditions. (A) Correlations between different samples. (B) Volcano plot of DEGs identified under freezing stress. Red dots represent up-regulated DEGs and blue dots represent down-regulated DEGs. (C) GO annotation results for the respective DEGs.
L170: How was the analysis of the MAPK signaling pathway done? Where are the results presented for the other genes? This question applied also to L175-177. All results should be shown in Suppl. Tables.
L173: responded to
L174: rephrase the sentence to “a large number of DEGs was assigned to the ABA signal transduction pathway”
Legend of Figure 2A: Bubble plot of the results of DEGs KEGG annotation.
L183: after freezing stress
Table S2: DAM Kegg annotation, in Table only 171 metabolic annotations, in text 353 are mentioned - please explain
L188: great similarity within samples of the same condition.
Fig. 3a and B. It is not explained if this figure derived from 10543 DAM or 353 metabolites.
L195: Did the authors really mean “enriched” or only “assigned to”? Check also L202 for the same, enrichment or assignment?
L207: connected pathways
L210: please provide a reference
L213: reference for epigallocatechin as important potential source of antioxidants is missing
L215: reference is missing
Fig. 3 The labelling of the axes is very small.
Fig. 4A The labelling is too small and hard to read.
L217, legend Fig 4A: Are these data derived from enrichment analysis or just DEG and DAM analysis – these are two different things.
L220/221: These are not all regulatory enzymes but also synthesizing enzymes, better say “respective enzymes”.
Gene expression of pathways is not possible, better say: expression of genes involved in pathways
L230: Phrasing of the sentence should be improved.
L234: Here it is not clear if the enrichment in photosynthesis was observed under CK or FS as the sentence before deals with CK.
L240: Rephrase the sentence for more clarity
L245: transduction
L248: Pyr/PYL- PP2C-SnRK2 contribute to stomata closure
L251: add reference
L255: add reference
L262-267: no literature cited for statements on lignin synthesis
L269-275: no references provided for statements
L277/278: The authors mentioned hormones and enzymes but did not investigate them in this manuscript.
L279: no figures for lignin provided
L282: observed
Author Response
Reply: All authors thank the reviewers for their summary of our paper. We really appreciate your help and patience. We have seriously thought about them and provided our response to reviewers. All the authors fully agree with what the Reviewer #2 are pointing out. We have efforted to correct the mistakes and improve the English of the manuscript. We further describe the methods. In addition, we further improved the expression of results and conclusions to make results clearer and conclusions more credible. For the detailed response, please see below.
The authors investigated 60-year old tea plants on transcriptomic and metabolomics level to get insights in the underlying freezing tolerance mechanisms. They mainly identified biosynthesis of secondary metabolites, as e.g. flavonoids, as main driver of freezing tolerance.
Reply: Thanks very much for this comment. In this study, transcriptome and metabolome analysis were performed for the first time using natural overwintering C.sinensis. 3,880 DEGs and 353 DAMs were obtained. KEGG annotation of DEGs and DAMs of freezing-stressed C.sinensis leaves showed that both sets of the data were significantly enriched in phenylpropanoid biosynthesis, flavone and flavonol biosynthesis and flavonoid biosynthesis (Figures 4A and 4B). 32 DAMs were intensively enriched in the three connection pathways, of which 31 DAMs were up-regulated and only 1 down-regulated. All the DAMs in flavone and flavonol biosynthesis and flavonoid biosynthesis pathways were upregulated. Flavone, flavonol and flavonoid biosynthesis were significantly up-regulated in freezing-stressed C.sinensis leaves, which played an essential role.
The language in the whole manuscript has to be intensively revised, especially in the discussion. Several sentences contain inaccuracies generated by language uncertainties.
Reply: Thanks a lot for your suggestion. We have efforted to correct the mistakes and improve the English of the manuscript. And Changes the uncertain languages to improve readability. Partial changes are as follows:
“Correlation analyses between samples and gene expression. (A) The analysis of correlation and difference between samples within and between groups. (B) Volcano plot of DEGs under freezing stress. Red dots represent up-regulated DEGs and blue dots represent down-regulated DEGs. (C) GO annotation results for DEGs.” changed to “Improve to: Correlation analyses between the gene expression of samples under control and freezing stress conditions. (A) Correlations between different samples. (B) Volcano plot of DEGs identified under freezing stress. Red dots represent up-regulated DEGs and blue dots represent down-regulated DEGs. (C) GO annotation results for the respective DEGs.”
“The ABA signal transduction had a large number of DEGs” changed to “A large number of DEGs was assigned to the ABA signal transduction pathway”.
“The results of bubble plot of DEGs KEGG annotation” changed to “Bubble plot of the results of DEGs KEGG annotation”.
“there was a great similarity within the groups” changed to “great similarity within samples of the same condition”.
“Gene expression of phenylpropanoid biosynthesis, flavone and flavonol biosynthesis and flavonoid biosynthesis pathways of freezing stress.” changed to “Expression of genes involved in phenylpropanoid biosynthesis, flavone and flavonol biosynthesis and flavonoid biosynthesis pathways of freezing stress.”
Samples were only taken in winter at subzero temperatures and in spring at warmer temperatures, but a sampling in fall before winter would have been desirable to analyze the initial status before freezing.
Reply: Thanks a lot for your suggestion. Camellia sinensis can grow regularly at 10-35 °C but nearly dead at -10 °C. -14°C is the lowest temperature of the whole year. Meanwhile, the C.sinensis was in a state of near-death growth. In order to find out how the C.sinensis recover from near-death state to normal growth, and the C.sinensis was sampled.
Sampling was only done in one year, even that for experiments under natural conditions a sampling over two to three years is necessary to detect general patterns even under changing weather conditions. Additionally, only a very low amount of samples was analyzed with three for transcriptomics and six for metabolomics.
Reply: Thanks very much for this comment. Tea leaves were sampled from other published articles, and each treatment of these transcriptome articles were repeated three times and each treatment of metabolome articles were repeated less than six replicates.
References
[1] Liang T, Yuan Z, Fu L, Zhu M, Luo X, Xu W, Yuan H, Zhu R, Hu Z, Wu X. Integrative Transcriptomic and Proteomic Analysis Reveals an Alternative Molecular Network of Glutamine Synthetase 2 Corresponding to Nitrogen Deficiency in Rice (Oryza sativa L.). Int J Mol Sci. 2021 Jul 18;22(14):7674
[2] Zhang X, Li F, Ding Y, Ma Q, Yi Y, Zhu M, Ding J, Li C, Guo W, Zhu X. Transcriptome Analysis of Two Near-Isogenic Lines with Different NUE under Normal Nitrogen Conditions in Wheat. Biology (Basel). 2021 Aug 17;10(8):787.
[3] Janz D, Behnke K, Schnitzler JP, Kanawati B, Schmitt-Kopplin P, Polle A. Pathway analysis of the transcriptome and metabolome of salt sensitive and tolerant poplar species reveals evolutionary adaption of stress tolerance mechanisms. BMC Plant Biol. 2010 Jul 17;10:150.
[4] Rothenberg DO, Yang H, Chen M, Zhang W, Zhang L. Metabolome and Transcriptome Sequencing Analysis Reveals Anthocyanin Metabolism in Pink Flowers of Anthocyanin-Rich Tea (Camellia sinensis). Molecules. 2019 Mar 18;24(6):1064.
In principle, the manuscript does not describe an integrative analysis (see headline) of transcriptomic and metabolomics data as for such an analysis data should be correlated together.
Reply: Thanks for your nice suggestion. Transcriptome data and metabolome data were correlated in section 5 (3.5. Integrative analysis of gene expression and metabolite levels) of the results. KEGG annotation of DEGs and DAMs of freezing-stressed C.sinensis leaves showed that both sets of the data were significantly enriched in phenylpropanoid biosynthesis, flavone and flavonol biosynthesis and flavonoid biosynthesis. 32 DAMs were intensively enriched in the three connection pathways, of which 31 DAMs were up-regulated and only 1 down-regulated. C.sinensis regulate freezing stress by regulating the expression of pyL and other enzymes in the pathways to affect the metabolism of flavone, flavonol, flavonoid and other substances.
The sampling needs to be described in more detail in the methods (time of the day, which leaves, how many leaves etc.).
Reply: Thanks a lot for your suggestion. We supplemented the methods with sampling time, leaves information. 2.1. Plant material and sampling content becomes “The experiment was carried out in the 60-year-old overwintering C.sinensis garden ((N 36°13′9″, E 117°3′16″) on Mount Tai. The first sampling (freezing stress) took place at 16:00 on 30 December 2020, with a minimum temperature of -14 °C and an average temperature of -10 °C as the lowest temperature of the year. At 16:00 on March 30, 2021, a second sample (control) was taken of the tea plants at an average temperature of 15°C. Tea trees with good growth status were selected for the first sampling, and the tea trees were marked after sampling, and the same tea tree was selected for the second sampling. These samples were immediately frozen in liquid nitrogen and stored at −80°C for subsequent sequencing. Twelve upper leaves with good growth status were sampled each time, of which nine were used for analysis and three were reserved. The experiment was divided into freezing stress group (FS) and control check group (CK). 9 samples were sampled from each group, of which 3 samples (FS_1, FS_2, FS_3, CK_1, CK_2, CK_3) were sequenced for transcriptome and 6 samples (FS-1, FS-2, FS-3, FS-4, FS-5, FS-6, CK-1, CK-2, CK-3, CK-4, CK-5, CK-6) were sequenced for metabolome.”
In the results section it is sometimes unclear if the authors just explain an assignment to certain pathways or really a statistical enrichment analysis, as often mentioned.
Reply: Thanks very much for this comment. We performed GO enrichment analysis and KEGG pathway enrichment analysis for all DEGs, and KEGG pathway enrichment analysis for all DAMs of samples. For the certain pathways, they are part of overall KEGG annotation results. KEGG annotates the results of all pathways, as well as the results of a certain pathway.
In the discussion, a lot of references are missing for the statement taken, for examples see detailed comments below. The discussion is in general quite poor, e.g. no references for the role of flavonoids in freezing tolerance are cited.
Reply: Thanks for your nice suggestion. We included references to previous research in the discussion.
References
[1] Wang Y, Xiong F, Nong S, Liao J, Xing A, Shen Q, Ma Y, Fang W, Zhu X. Effects of nitric oxide on the GABA, polyamines, and proline in tea (Camellia sinensis) roots under cold stress. Sci Rep. 2020 Jul 22;10(1):12240.
[2] Liu Q, Luo L, Zheng L. Lignins: Biosynthesis and Biological Functions in Plants. Int J Mol Sci. 2018 Jan 24;19(2):335.
[3] Wang W, Li Y, Dang P, Zhao S, Lai D, Zhou L. Rice Secondary Metabolites: Structures, Roles, Biosynthesis, and Metabolic Regulation. Molecules. 2018 Nov 27;23(12):3098.
[4] Böttner L, Grabe V, Gablenz S, Böhme N, Appenroth KJ, Gershenzon J, Huber M. Differential localization of flavonoid glucosides in an aquatic plant implicates different functions under abiotic stress. Plant Cell Environ. 2021 Mar;44(3):900-914.
[5] Knez Hrnčič M, Ivanovski M, Cör D, Knez Ž. Chia Seeds (Salvia hispanica L.): An Overview-Phytochemical Profile, Isolation Methods, and Application. Molecules. 2019 Dec 18;25(1):11.
In the conclusion, statements were drawn on hormones and enzymes even that the authors did not measure them.
Reply: Thanks very much for this comment. In the fifth part of the results, the expression of genes regulating enzymes between metabolites in the pathway is visualized in Figures 4B. In addition, hormones were deleted from the conclusion.
Fig. 5 is not very helpful, as it is too general. The first part was not investigated at all by the authors, fatty acid was not mentioned at all and lignin only shortly in the whole manuscript and the physiological response mentioned in the green box was not shown at all.
Reply: Thanks a lot for your suggestion. According to your suggestion, we have deleted Figure 5. In addition, we reorganized the conclusions and changed them to" In conclusion, the signals sensed by the freezing stress receptors of C.sinensis under freezing stress were transduced and transmitted through hormones, etc., and the C.sinensis that received the signals were metabolized by lignin, flavonoids, carbohydrates and other substances to improve stress resistance. In addition, stomatal closure might be promoted by ABA signaling to resist freezing stress. In this experiment, a large number of DEGs, metabolites and key pathways were observed from C.sinensis under freezing stress. It is worth further studying how the freezing stress signals are conducted and how these significantly related candidate genes and metabolites regulate the freezing resistance of C.sinensis. The results laid a foundation for improving the freezing resistance of C.sinensis by genetic engineering.".
A detailed author contribution is missing at the end of the manuscript.
Reply: Thanks very much for this comment. We have added authors' contributions to the newly submitted manuscript.
Further comments:
Reply: We really appreciate your help and patience. We have revised your questions one by one. For the detailed response, please see below.
L25: please rephrase as pathways cannot have a concentration
Reply: “concentration” changed to “enrichment”.
L42: When plants are subjected to freezing stress, photosynthesis will be inhibited and cell structure destroyed, resulting in
Reply: “When plants are subjected to freezing stress, they will inhibit photosynthesis, destroy cell structure, resulting in” changed to “When plants are subjected to freezing stress, photosynthesis will be inhibited and cell structure destroyed, resulting in”.
L49: After being freeze stressed, the photosynthesis of F. cirrhosa would return to the normal level – the statement should be checked, as photosynthesis is inhibited after freezing stress as stated in L42
Reply: “After being freezed stress” changed to “After the relief of freezing stress”.
L58: „were analyzed“ as the metabolome cannot be sequenced
Reply: “sequenced” changed to “analyzed”.
L60-62: Flavonoids are not known for their regulatory effects.
Reply: Reference have been added.
[1] Sharma A, Shahzad B, Rehman A, Bhardwaj R, Landi M, Zheng B. Response of Phenylpropanoid Pathway and the Role of Polyphenols in Plants under Abiotic Stress. Molecules. 2019 Jul 4;24(13):2452.
L72-74: delete “for 60 years” as this implied that study was done over 60 years
Reply: “for 60 years” has been deleted.
L74: here “gene regulation and expression” describe similar things
Reply: “gene regulation and expression” changed to “gene regulation”.
L75/76: Only the metabolic status, not the pathways were analyzed. Only conclusions were drawn on metabolic pathways.
Reply: KEGG annotation of DEGs and DAMs of freezing-stressed C.sinensis leaves showed that both sets of the data were significantly enriched in phenylpropanoid biosynthesis, flavone and flavonol biosynthesis and flavonoid biosynthesis (Figures 4A and 4B).
L77: protein level was not analyzed
Reply: “protein” changed to “metabolite”.
L94: investigated for metabolome analysis
Reply: “sequenced” changed to “analyzed”.
L97-99: The methods should be written in the respective order, first preparation of RNA samples and the libraries, then sequencing.
Reply: “Using the Truseq RNA sample preparation kit, the mRNA library was constructed and sequenced in depth, and the RNA sample was prepared according to the manufacturer’s protocol. RNA quality was evaluated by agarose gel electrophoresis and OD260/230 ratio, and the cDNA library was constructed after the samples were qualified. The library were qualified by Agilent 2100 bioanalyzer and quantified by Qubit and qPCR” changed to “Using the Truseq RNA sample preparation kit, and the RNA sample was prepared according to the manufacturer’s protocol. RNA quality was evaluated by agarose gel electrophoresis and OD260/230 ratio, and the cDNA library was constructed after the samples were qualified. The library were qualified by Agilent 2100 bioanalyzer and quantified by Qubit and qPCR”.
L106/107: check sentence
Reply: “Using Hisat2 to map clean data to reference genome "shuchazao" (http://tpia.teaplant.org/download.html), and statistical mapping information” changed to “Using Hisat2 to map the clean data to "shuchazao" reference genome (http://tpia.teaplant.org/download.html), and calculate mapping information”.
L118: 100 l – this amount seems to be very high, please check
Reply: “100 l” changed to “100 µl”.
L127: quality control
Reply: “Quanlity Contral” changed to “quality control”.
L145-146: Sentence should be checked and an explanation ahould be given for different samples quality
Reply: This is explained in the second paragraph of the discussion. “In transcriptome sequencing, the gene expression of FS was smaller than CK, and the number of down-regulated DEGs was more than up-regulation. Many of these down-regulated genes were associated with normal development and physiological metabolism of C.sinensis, suggesting that at 10°C, and C.sinensis could recover from their epistotic dormancy state to normal vegetative growth.”
L156-161: No statistical gene enrichment analysis was done, so avoid the term enrichment, rather mention that DEG were assigned to the respective GO categories
Reply: We performed GO enrichment analysis and KEGG pathway enrichment analysis for all DEGs, and KEGG pathway enrichment analysis for all DAMs of samples. For the certain pathways, they are part of overall KEGG annotation results. KEGG annotates the results of all pathways, as well as the results of a certain pathway.
Figure 1. Improve to: Correlation analyses between the gene expression of samples under control and freezing stress conditions. (A) Correlations between different samples. (B) Volcano plot of DEGs identified under freezing stress. Red dots represent up-regulated DEGs and blue dots represent down-regulated DEGs. (C) GO annotation results for the respective DEGs.
Reply: “Correlation analyses between samples and gene expression. (A) The analysis of correlation and difference between samples within and between groups. (B) Volcano plot of DEGs under freezing stress. Red dots represent up-regulated DEGs and blue dots represent down-regulated DEGs. (C) GO annotation results for DEGs.” changed to “Improve to: Correlation analyses between the gene expression of samples under control and freezing stress conditions. (A) Correlations between different samples. (B) Volcano plot of DEGs identified under freezing stress. Red dots represent up-regulated DEGs and blue dots represent down-regulated DEGs. (C) GO annotation results for the respective DEGs.”
L170: How was the analysis of the MAPK signaling pathway done? Where are the results presented for the other genes? This question applied also to L175-177. All results should be shown in Suppl. Tables.
Reply: We added the whole transcriptomic data and provided details of all DEGs in Table S2.
L173: responded to
Reply: “response to” changed to “responded to”.
L174: rephrase the sentence to “a large number of DEGs was assigned to the ABA signal transduction pathway”
Reply: “The ABA signal transduction had a large number of DEGs” changed to “A large number of DEGs was assigned to the ABA signal transduction pathway”.
Legend of Figure 2A: Bubble plot of the results of DEGs KEGG annotation.
Reply: “The results of bubble plot of DEGs KEGG annotation” changed to “Bubble plot of the results of DEGs KEGG annotation”.
L183: after freezing stress
Reply: “of freezing stress” changed to “after freezing stress”.
Table S2: DAM Kegg annotation, in Table only 171 metabolic annotations, in text 353 are mentioned - please explain
Reply: KEGG annotation was carried out for all metabolites detected in untargeted metabolomics, and 353 DAMs and 171 metabolic pathways were annotated. And each metabolic pathway has multiple DAMs.
L188: great similarity within samples of the same condition.
Reply: “there was a great similarity within the groups” changed to “great similarity within samples of the same condition”.
Fig. 3a and B. It is not explained if this figure derived from 10543 DAM or 353 metabolites.
Reply: “Principal component analysis (PCA) showed that the two treatments were significantly differentiated for the first component (79.3% for PC1) (Figure 3 A). The OPLS-DA showed a clearer distinction between FS and CK group of samples (Figure 3 B).” changed to “Principal component analysis (PCA) of the 10543 DAMs showed that the two treatments were significantly differentiated for the first component (79.3% for PC1) (Figure 3 A). The OPLS-DA of the 10543 DAMs showed a clearer distinction between FS and CK group of samples (Figure 3 B)”.
L195: Did the authors really mean “enriched” or only “assigned to”? Check also L202 for the same, enrichment or assignment?
Reply: We performed GO enrichment analysis and KEGG pathway enrichment analysis for all DEGs, and KEGG pathway enrichment analysis for all DAMs of samples. For the certain pathways, they are part of overall KEGG annotation results. KEGG annotates the results of all pathways, as well as the results of a certain pathway. A single pathway can also be called enrichment.
References
[1] Huang J, Zhao X, Chory J. The Arabidopsis Transcriptome Responds Specifically and Dynamically to High Light Stress. Cell Rep. 2019 Dec 17;29(12):4186-4199.e3.
[2] Fox H, Doron-Faigenboim A, Kelly G, Bourstein R, Attia Z, Zhou J, Moshe Y, Moshelion M, David-Schwartz R. Transcriptome analysis of Pinus halepensis under drought stress and during recovery. Tree Physiol. 2018 Mar 1;38(3):423-441.
L207: connected pathways
Reply: “connection pathways” changed to “connected pathways”.
L210: please provide a reference
Reply: Reference have been added.
[1] Vogt T. Phenylpropanoid biosynthesis. Mol Plant. 2010 Jan;3(1):2-20.
L213: reference for epigallocatechin as important potential source of antioxidants is missing
Reply: Reference have been added.
[1] Ambigaipalan P, Oh WY, Shahidi F. Epigallocatechin (EGC) esters as potential sources of antioxidants. Food Chem. 2020 Mar 30;309:125609.
L215: reference is missing
Reply: Reference have been added.
[1] Wen W, Alseekh S, Fernie AR. Conservation and diversification of flavonoid metabolism in the plant kingdom. Curr Opin Plant Biol. 2020 Jun;55:100-108.
Fig. 3 The labelling of the axes is very small.
Reply: Fig. 3 has been modified in the new manuscript.
Fig. 4A The labelling is too small and hard to read.
Reply: Fig. 4 has been modified in the new manuscript.
L217, legend Fig 4A: Are these data derived from enrichment analysis or just DEG and DAM analysis – these are two different things.
Reply: Fig 4A was derived from kegg enrichment analysis of all DEGs and DAMs.
L220/221: These are not all regulatory enzymes but also synthesizing enzymes, better say “respective enzymes”.
Reply: “enzymes” changed to “respective enzymes”.
Gene expression of pathways is not possible, better say: expression of genes involved in pathways
Reply: “Gene expression of phenylpropanoid biosynthesis, flavone and flavonol biosynthesis and flavonoid biosynthesis pathways of freezing stress.” changed to “Expression of genes involved in phenylpropanoid biosynthesis, flavone and flavonol biosynthesis and flavonoid biosynthesis pathways of freezing stress.”
L230: Phrasing of the sentence should be improved.
Reply: “Through transcriptome sequencing, the gene expression level of FS was smaller than that of CK, and the number of down-regulated DEGs was more than up-regulation.” changed to “In transcriptome sequencing, the gene expression of FS was smaller than CK, and the number of down-regulated DEGs was more than up-regulation.”
L234: Here it is not clear if the enrichment in photosynthesis was observed under CK or FS as the sentence before deals with CK.
Reply: Through GO enrichment analysis of DEGs, DEGs were found to be enriched in photosynthesis (Figure 1C).
L240: Rephrase the sentence for more clarity
Reply: “Moreover, enriched biological processes (e.g., response to stimulus) and molecular functions (e.g., antioxidant activity and signal transducer activity) were related to abiotic stresses and inseparable from the membrane.” changed to “Moreover, the biological processes (e.g., response to stimulus) and molecular functions (e.g., antioxidant activity and signal transducer activity) of GO enrichment were related to abiotic stresses and inseparable from the membrane.”
L245: transduction
Reply: “transducation” changed to “transduction”.
L248: Pyr/PYL- PP2C-SnRK2 contribute to stomata closure
Reply: “Pyr/PYL-PP2C-SnRK2 could also close the stomata to improve the water holding capacity and freezing resistance of C.sinensis.” changed to “Pyr/PYL-PP2C-SnRK2 contribute to stomata closure to improve the water holding capacity and freezing resistance of C.sinensis.”.
L251: add reference
Reply: Reference have been added.
[1] Rubio S, Noriega X, Pérez FJ. Abscisic acid (ABA) and low temperatures synergistically increase the expression of CBF/DREB1 transcription factors and cold-hardiness in grapevine dormant buds. Ann Bot. 2019 Mar 14;123(4):681-689.
L255: add reference
Reply: Reference have been added.
[1] Wang Y, Xiong F, Nong S, Liao J, Xing A, Shen Q, Ma Y, Fang W, Zhu X. Effects of nitric oxide on the GABA, polyamines, and proline in tea (Camellia sinensis) roots under cold stress. Sci Rep. 2020 Jul 22;10(1):12240.
L262-267: no literature cited for statements on lignin synthesis
Reply: Reference have been added.
[1] Liu Q, Luo L, Zheng L. Lignins: Biosynthesis and Biological Functions in Plants. Int J Mol Sci. 2018 Jan 24;19(2):335.
L269-275: no references provided for statements
Reply: References have been added.
[1] Wang W, Li Y, Dang P, Zhao S, Lai D, Zhou L. Rice Secondary Metabolites: Structures, Roles, Biosynthesis, and Metabolic Regulation. Molecules. 2018 Nov 27;23(12):3098.
[2] Böttner L, Grabe V, Gablenz S, Böhme N, Appenroth KJ, Gershenzon J, Huber M. Differential localization of flavonoid glucosides in an aquatic plant implicates different functions under abiotic stress. Plant Cell Environ. 2021 Mar;44(3):900-914.
[3] Knez Hrnčič M, Ivanovski M, Cör D, Knez Ž. Chia Seeds (Salvia hispanica L.): An Overview-Phytochemical Profile, Isolation Methods, and Application. Molecules. 2019 Dec 18;25(1):11.
L277/278: The authors mentioned hormones and enzymes but did not investigate them in this manuscript.
Reply: In the fifth part of the results, the expression of genes regulating enzymes between metabolites in the pathway is visualized in Figures 4B. In addition, hormones were deleted from the conclusion.
L279: no figures for lignin provided
Reply: According to your suggestion, we have deleted Figure 5.
L282: observed
Reply: “obtained” changed to “observed”.
Round 2
Reviewer 1 Report
The authors have made sufficient changes in the manuscript to make the conclusion more credible. From my point of view theses changes help to improve the manuscript but do not resolve the issue of the novelty of the study. When it is great that this study has been made in C. sinensis (a non-model plant), it responses to freezing seem to be similar to other plants responses. However, the results are well presented (and point out that these responses are common among different plants) and the conclusions have been improved (but it could be better) to explain the potential mechanisms that C. sinensis activates to cope with freeze stress.
Reviewer 2 Report
The authors considered most of the reviewer suggestions. Nevertheless, there are still things, which have to be clarified or have to be changed by the authors. I directly answer to the author replies.
Reply: Thanks for your nice suggestion. Transcriptome data and metabolome data were correlated in section 5 (3.5. Integrative analysis of gene expression and metabolite levels) of the results.
The definition of an integrated analysis requires a mathematical correlation between DEGs and DAMs. It is not sufficient to analyze the data separately and to draw then conclusions on pathways which have been increased for DEG as well as DAM. As this was done in your manuscript you should at least avoid the word “correlated” in the respective chapter as this is reserved for a mathematical correlation. “Integrated” means always a direct correlation of the data.
Unfortunately, recently the term integrated is used wrongly also in a lot of other publications.
Reply: Thanks very much for this comment. In the fifth part of the results, the expression of genes regulating enzymes between metabolites in the pathway is visualized in Figures 4B. In addition, hormones were deleted from the conclusion.
I understood that that the expression of genes was shown in Figure 4. When you talk about the genes you should not use the word “enzyme” as you did not measure enzyme, as you did not measure enzyme activity. Instead of using the term “enzyme” you have to say “genes encoding enzymes”.
Reply: In addition, we reorganized the conclusions and changed them to" In conclusion, the signals sensed by the freezing stress receptors of C.sinensis under freezing stress were transduced and transmitted through enzymes, etc., and the C.sinensis that received the signals were metabolized by lignin, flavonoids, carbohydrates and other substances to improve stress resistance. In addition, stomatal closure might be promoted by ABA signaling to resist freezing stress. In this experiment, a large number of DEGs, metabolites and key pathways were observed from C.sinensis under freezing stress. It is worth further studying how the freezing stress signals are conducted and how these significantly related candidate genes and metabolites regulate the freezing resistance of C.sinensis. The results laid a foundation for improving the freezing resistance of C.sinensis by genetic engineering."
The conclusion has to be rephrased again as it still contains a lot of hypothesis. The authors should stick to their own results and do not mention things they did not investigate as “signals sensed” or “freezing stress receptors” or “stomatal closure”. The authors did not provide any data on that. The first and second sentence should be deleted and new sentences on their own results should be provided. Also, the meaning of the first sentence is not clear as C. sinensis cannot be metabolized.
Reply to L60-62: Flavonoids are not known for their regulatory effects. Reference have been added.
[1] Sharma A, Shahzad B, Rehman A, Bhardwaj R, Landi M, Zheng B. Response of Phenylpropanoid Pathway and the Role of Polyphenols in Plants under Abiotic Stress. Molecules. 2019 Jul 4;24(13):2452.
I am aware of the references on flavonoids. Accumulation of flavonoids and increase of expression of genes encoding flavonoid biosynthesis related enzymes was reported as stress response in previous publications. But, flavonoids themselves do not regulate the stress response. Be careful with the term “regulator”. Also, the provided reference do not provide any evidence for a regulatory role.
Reply to L145-146: Sentence should be checked and an explanation ahould be given for different sample quality:
This is explained in the second paragraph of the discussion. “In transcriptome sequencing, the gene expression of FS was smaller than CK, and the number of down-regulated DEGs was more than up-regulation. Many of these down-regulated genes were associated with normal development and physiological metabolism of C.sinensis, suggesting that at 10°C, and C.sinensis could recover from their epistotic dormancy state to normal vegetative growth.”
The respective sentence needs to be rephrased to: In addition, the total exon length, average transcript length and N50 length (without intron) in samples of the FS group were also smaller than in the CK group.
The sentence in the discussion should be rephrased to: “In transcriptome sequencing, the transcript number of FS samples was smaller than that of CK samples, and the number of down-regulated DEGs was higher than the number of up-regulated DEGs. Many of these down-regulated genes were associated with normal development and physiological metabolism of C. sinensis, suggesting that at 10°C C. sinensis recovered from the dormancy state to normal vegetative growth.”
Reply to Table S2: DAM Kegg annotation, in Table only 171 metabolic annotations, in text 353 are mentioned - please explain.
KEGG annotation was carried out for all metabolites detected in untargeted metabolomics, and 353 DAMs and 171 metabolic pathways were annotated. And each metabolic pathway has multiple DAMs.
Then you have to rephrase the respective sentence to: A total of 101543 DAMs were obtained… and 353 metabolites were assigned to 171 KEGG functional categories.
Furthermore, the role of flavonoids for freezing tolerance previously described was not discussed. Possible references might be: Schulz et al. 2016 (doi: 10.1038/srep34027) and Zhao et al. 2019 (https://doi.org/10.3389/fpls.2019.01675).
